# Prevention of Epidural Catheter Migration and Inflammation by Tunneling: A Systematic Review and Meta-Analysis

**DOI:** 10.3390/jcm14165788

**Published:** 2025-08-15

**Authors:** Merel N. van Kassel, Jeroen Hermanides, Philipp Lirk, Markus W. Hollmann

**Affiliations:** 1Departement of Anesthesiolgy, Amsterdam University Medical Center, 1100 DD Amsterdam, The Netherlands; m.n.vankassel@amsterdamumc.nl (M.N.v.K.); j.hermanides@amsterdamumc.nl (J.H.); 2Brigham and Women’s Hospital, Boston, MA 02115, USA; philipp.lirk@uth.tmc.edu

**Keywords:** epidural anesthesia, epidural analgesia, tunneling technique, catheter migration, inflammation

## Abstract

**Background**/**Objectives**: This systematic review and meta-analysis evaluated the efficacy of tunneling as a fixation technique of epidural catheters on catheter migration and infection compared to conventional fixation techniques. **Methods**: All studies comparing any epidural tunnel fixation technique with any conventional epidural fixation technique (any technique, e.g., adhesive tape) were included, with the exclusion of the obstetric population and the exclusion of caudal anesthesia. MEDLINE and Embase were searched on 18 April 2025 to identify these studies. To evaluate the risk of bias in the selected studies, the Risk of Bias assessment tool from the Cochrane Handbook was used. Risk ratios (RRs) with 95% confidence intervals (CIs) were determined to describe the difference in migration and infection between the two groups. **Results**: Eleven studies, with a total number of 23,695 patients, were included in this systematic review. Pooled data of the nine studies reporting data on migration showed that tunneling reduced the incidence of inward migration (RR, 0.33; 95% CI, 0.19 to 0.55; n = 613; *p* < 0.0001; I2 = 0%) but not outward migration (RR, 0.60; 95% CI, 0.25 to 1.43; n = 745; *p* = 0.25; I2 = 77%). Pooled data from the nine studies on the incidence of infection revealed no statistically significant group difference. **Conclusions**: Tunneling of the epidural catheter as a fixation technique reduces the risk of inward migration compared to conventional fixation methods. However, for outward migration, tunneling does not reduce the risk of infection. There is a large heterogeneity in fixation methods described in the literature. In summary, this first comprehensive systematic review and meta-analysis corroborates only part of the benefits widely ascribed to epidural catheter tunneling.

## 1. Introduction

Epidural analgesia can be very effective in the management of post-operative pain, with high levels of patient satisfaction [1]. In daily clinical practice, however, epidural catheters are associated with a relatively high secondary failure rate due to catheter migration and the potential for complications, including epidural hematoma and abscess [2].

The incidence of epidural catheter migration varies widely in the literature, with incidences reported between 5% and 50% [3,4,5,6]. Catheter migration can lead to loss of analgesic effect, especially when migrating outward or laterally (transforaminal escape), and to potential complications, such as intravascular, subdural, or subarachnoid injection of local anesthetic, if migrating is directed inwards [7,8]. The incidence of infection of an epidural catheter is estimated between 0.8 and 5.5%, depending on the definition [9,10]. The most common route of bacterial pathogen migration is along the cutaneous track of the epidural catheter [3,11].

To alleviate both catheter migration and risk of infection, tunneling of epidural catheters has been proposed to allow for a better fixation, thereby reducing catheter movement underneath the skin and thus minimizing bacterial movement and colonization along the catheter [3,7,8,9]. This systematic review was conducted to evaluate the effect of tunneling epidural catheters on catheter migration and infection by comparing tunneling with conventional fixation techniques (any technique, e.g., adhesive tape). In addition, it was determined whether a difference in outward migration leads to a difference in analgetic effectiveness or patient satisfaction.

## 2. Materials and Methods

The study presented here is in line with the guidelines for Systematic Reviews and Meta-Analysis of the PRISMA Statement paper 2020 [12]. The PICO framework, search strategy, and data extraction form were prepared in advance; for details, see the Appendix A. The study protocol was registered and published Systematic Review in the PROSPERO database on 17 April 2025 (available online https://www.crd.york.ac.uk/PROSPERO/view/CRD420251032567).

### 2.1. Search Strategy

A systematic search was performed on 18 April 2025 in two medical databases: MEDLINE (via PubMed) and Embase. A predefined search query was used in this search, including MeSH (Medical Subject Headings) terms; see the full search in the Appendix A. There was no filter applied, and there were no language restrictions. EndNote (EndNote, Version 21.2) was used for the management of the identified records and the selection process. After removing the duplicates both automatically and manually, two independent authors (M.N.v.K. and M.W.H.) screened the records for eligibility based on the title/abstract and by full text reading. A third independent author (J.H.) could be consulted in case of possible differences between the decisions of two authors in the screening process.

### 2.2. Study Selection

All studies comparing any epidural tunnel fixation technique (tunneled group, TG) with any conventional epidural fixation technique (control group, CG) in adult patients were included in this research. The PICO framework is provided in the Appendix A.

### 2.3. Inclusion and Exclusion Criteria

All studies that compared at least two groups of patients, those who underwent epidural fixation by any tunneling technique and those who underwent a conventional fixation method (any technique, e.g., adhesive tape), were included. All types of studies were included in order not to miss important outcomes of, for example, large registry studies. Studies without original study data, case reports, cohort studies without a comparison group, or studies published only in abstract form were excluded. Also, articles investigating other types of catheters or techniques, such as central venous catheters, caudal catheters, or regional nerve blocks, were excluded.

### 2.4. Outcomes

The primary outcomes were between-group differences in the incidence of migration (in- and outward) between the tunneled epidural and the conventional fixation group and signs of infection, as defined by the authors.

All definitions of infection and its precursors, inflammation and colonization, as specified by the authors, were included in this systematic review. For migration, all studies that reported any form of migration, as described by the authors, were included in this systematic review. However, the most commonly used definition in the literature is >2 cm from the original position for outward movement and >1 cm for inward movement; therefore, a subgroup analysis was performed for these specific definitions, defined as “significant migration” [13,14]. The secondary outcome was the difference in analgesia or patient satisfaction between the groups. However, the variation in definitions and measurements of adequate analgesia and/or patient satisfaction is very broad; therefore, this outcome will only be described.

### 2.5. Data Extraction

Two authors (M.N.v.K. and M.W.H.) assessed every article’s title, abstract, and full text independently and evaluated whether it met the inclusion criteria. A third independent author (J.H.) was selected to decide about possible differences between the decisions of the two authors in this process. Data were transferred to Excel (Microsoft^®^ Excel^®^, Microsoft 365 MSO. Version 2402); see Appendix A. The following data were extracted from the included articles: the first author, year of publication, DOI, country, study design, patient demographics, fixation techniques, duration of epidural in situ, level of epidural insertion, and the data and parameters for the predefined outcomes.

### 2.6. Data Synthesis and Analysis

For the dichotomous outcomes of migration and infection, risk ratios (RRs) with 95% confidence intervals (CIs) were determined to describe the difference between the two groups (tunneled vs. conventional fixation methods). If raw data were not available, the corresponding author was contacted. Meta-analyses were performed using RevMan Review Manager (RevMan) software, Version 5.4, The Cochrane Collaboration.

A random-effects model was utilized to account for clinical and methodological heterogeneity across studies. Statistical heterogeneity was assessed using I2 statistics and the Q test. I2 values ranging from 30% to 60% were categorized as moderate heterogeneity, while values between 50% and 90% indicated substantial heterogeneity. Statistical significance was defined as *p* < 0.05. Because of the high heterogeneity of fixation techniques, a subgroup analysis was performed on the difference in incidence of in- and outward migration in tunneling vs. conventional fixation with adhesive tape only (i.e., no Lock-It device or suturing). For the pooled results, the Mantel–Haenszel method was applied.

### 2.7. Risk of Bias

Two authors (M.N.v.K. and M.W.H.) independently evaluated the risk of bias in the selected studies using the “Risk of Bias” assessment tool (ROB-2) from the Cochrane Handbook [15]. In case of any disagreements, consensus would be sought in discussion with each other to achieve agreement on which category it should be classified. A summary figure of the risk of bias was created by using the ROB-2 tool, a revised tool for assessing risk of bias in randomized trials [16]. For the overall risk of bias assessment, the selected studies were categorized as follows: Those considered to have a low risk of bias across all domains were classified as “low risk”; studies raising some concerns in more than one domain without any high risk of bias were classified as “some concerns”; and studies assessed to have a high risk of bias or in more than one domain or “no information” and/or “some concerns” ≥ 3 domains were classified as “high risk”.

### 2.8. Assessments of Confidence

The quality of the included studies was assessed with the use of the Grading of Recommendations Assessment, Development, and Evaluation (GRADE) system, based on the recommendations of the Cochrane Collaboration. Following this assessment, a GRADE evidence profile table was generated utilizing GRADEpro software: GRADEpro GDT: GRADEpro Guideline Development Tool [Software]. McMaster University and Evidence Prime, 2025 (available online: https://www.gradepro.org/ (accessed on 19 April 2025)) to categorize all outcomes as very low, low, moderate, or high quality.

## 3. Results

### 3.1. Search Results and Study Characteristics

A total of 156 papers were identified through a search of PubMed, and 302 papers were identified through a search in EMBASE. Initially, 148 duplicate studies were excluded, leaving 310 studies for further assessment. Of those, 285 articles were excluded based on title and abstract, leaving 25 for full-text reading. Ten articles met the inclusion criteria and were included in the final analysis [3,8,9,13,14,17,18,19,20,21]. Figure 1 provides a summary of the flowchart search and selection process. The characteristics of the ten included studies, with a total number of 23,645 patients for this systematic review, are summarized in Table 1. In Table 2, the definitions of migration, infection, and analgetic failure described in each article are provided.

### 3.2. Primary Outcomes

#### 3.2.1. Migration

When comparing the group with tunneling as a fixation technique and the group with a conventional method of fixation (for methods, see Table 1), the combined analysis of all eight studies [3,8,13,14,18,19,20,21] reporting data on this outcome revealed a statistically significant difference in migration (RR, 0.54; 95% CI, 0.33 to 0.89; n = 984; *p* = 0.02; I2 = 59%, Figure 2a).

Pooling data on in- and outward migration separately, both showed a statistically significant difference (inward migration RR, 0.35; 95% CI, 0.21 to 0.59; n = 1034; *p* < 0.0001; I2 = 0%, Figure 2b; and outward migration RR, 0.57; 95% CI, 0.34 to 0.95; n = 1234; *p* = 0.03; I2 = 54%, Figure 2c).

However, when only studies in the control group that used tape adhesive fixation were included, a statistically significant difference in the incidence of inward migration was found (RR, 0.33; 95% CI, 0.19 to 0.55; n = 613; *p* < 0.0001; I2 = 0%, Figure 3a) but not for outward migration (RR, 0.60; 95% CI, 0.25 to 1.43; n = 745; *p* = 0.25; I2 = 77%, Figure 3b).

Pooling the data of the three studies with data on significant migration (inward migration > 1 cm or outward migration > 2 cm), both inward (RR, 0.30; 95% CI, 0.13 to 0.69; n = 395; *p* = 0.005; I2 = 0%, Figure 4b) and outward migration (RR, 0.36; 95% CI, 0.20 to 0.65; n = 395; *p* = 0.0009; I2 = 0%, Figure 4c) revealed a statistically significant difference in favor of tunneling (both in- and outward migration (RR, 0.32; 95% CI, 0.20 to 0.52; n = 395; *p* < 0.00001; I2 = 0%, Figure 4a)) [13,18,21].

#### 3.2.2. Infection

The combined analysis of all nine studies [3,8,9,13,14,19,20,21] reporting data on any form of infection revealed no statistically significant difference (RR, 0.96; 95% CI, 0.59 to 1.57; n = 23,356; *p* = 0.05; I2 = 51%, Figure 5).

### 3.3. Secondary Outcome

Eight studies [3,8,9,13,14,17,19,21] reported outcomes on analgesia, pain scores, or patient satisfaction (Table 3). Five studies [13,14,17,19,21] reported no difference in analgesic effect and/or pain scores between de groups, and only one study [9] did find a significant difference in pain scores, with less pain in rest and during activity at 24 h in the tunneled group (*p* < 0.001). Two studies [9,21] reported a significant difference in overall patient satisfaction in favor of the tunneled group (both *p* < 0.001). One study [8] investigated patients’ comfort during epidural fixation and found that 77% of the patients disliked the tunneling technique during the procedure, compared to 0% of the patients in the control group.

### 3.4. Risk of Bias

The majority of the studies were classified as “low risk of bias” (n = 7); although there were some concerns in six of them, one study was classified overall as “some concerns” and one with “high risk of bias”. The risk of bias, explanation of the concerns, and a table outlining a summary of the findings of the quality assessment of the included studies can be found in the Appendix A.

### 3.5. Assessment of Confidence

The quality of evidence was assessed using the GRADE assessment to provide an overview of both the quality of evidence and the findings; see Appendix A. Due to heterogeneity in fixation methods, the differences in the definitions of migration and infection, and the inability to blind the participants, both outcomes were classified as low quality.

## 4. Discussion

The aim of this systematic review and meta-analysis was to evaluate the effect of tunneling epidural catheters as a fixation technique on catheter migration and signs of infection by comparing fixation by tunneling with conventional fixation techniques. It was found that tunneling does have a positive effect on reducing inward migration, although this is not clear for outward migration. Furthermore, tunneling does not seem to reduce signs of infection. In addition, there was no difference in analgesia, epidural failure, or pain scores between the two techniques.

Although a significant effect of tunneling on moderate inward migration and possibly also outward migration of the epidural catheter was found, we question the clinical relevance of this outcome. In the subgroup analyses on significant inward migration, a strict definition of inward migration (>1 cm) was used because of the theoretically more severe complications of inward migration, like intravascular, subdural, or subarachnoid injection of local anesthetic [7,8]. However, in this systematic review and meta-analysis, for the 1034 included patients in whom this outcome was reported, none of these complications were listed, including patients in the control group. Even if inward migration results in the above-described complications, the number needed to treat will be very high, as the complications are very rare; for example, the incidence of subarachnoid catheter movement is estimated at around 0.1–0.2% [13].

Furthermore, the highest risk of outward migration is analgesic failure due to migration of the catheter or one of the orifices out of the epidural space. With the use of multiorifice epidural catheters and the recommended practice of at least 4 cm insertion of the catheter into the epidural space, the 2 cm criterion is the minimum displacement required to extend the proximal orifice beyond the epidural space, which may cause analgesic failure [4,22]. The clinical relevance may be subordinate, since there was no difference between the two groups in epidural failure rate or pain scores. In fact, two studies described very precisely both patients with epidural failure without any significant movement of the catheter and patients with >4 cm migration who were satisfied with analgesia [3,19]. Overall, two studies reported a significant difference in patient satisfaction in favor of the tunneled group, though this must be weighed against the studies that show that the tunneling procedure itself is more uncomfortable for the patients [7,8].

The most common route of bacterial pathogen migration is contamination of the exit site and subsequent spread along the catheter track [3,9,11]. It has been suggested that tunneling allows for better fixation, which in turn reduces catheter movement underneath the skin, thus minimizing bacterial movement and colonization along the catheter [3,7]. However, based on the results of this meta-analysis, tunneling did not have a significant effect on signs of infection of the epidural catheter. Again, the heterogeneity for the tunneling techniques employed, for example, with or without a loop or suturing, could have influenced the results. Leaving a loop between the epidural and tunneling entrance, as a strategy to inhibit outward traction of the catheter, may actually promote bacterial contamination due to multiple skin punctures [8,9,20], whereas tunneling without a loop will move the skin entry of the catheter away from the site entry to the epidural space and could induce an additional barrier to bacterial infection [3]. This seems to be supported in a subgroup analysis on tunneling with or without a loop, which was outside the scope of this systematic review (see Appendix A). The risk of infection seems to be lower in the “tunneling with loop” group and higher in the “tunneling without loop” group, compared to the conventional method. Suturing the catheter for extra fixation, which was performed in the tunneling group of the study by Sellman et al., and in the control group in the study by Chadwick et al., might have caused more erythema [14,19,22]. Note that if tunneling may indeed reduce the rate of infection, as stated by the results of the large registry (n = 22,411) of Bomberg et al., it has a high number needed to treat (NNT): NNT of 152 for mild infections and NNT of 257 for moderate infections [9]. In the context of infection, we also note that we excluded caudal epidural catheters, which feature their own set of considerations around infection control and tunneling.

This systematic review and meta-analysis have several limitations. Firstly, there was a large heterogeneity in the fixation methods of the included studies. Within the tunneling group as well as the conventional method group, different types of techniques were used. It can be assumed that different methods influence the results or make comparison more difficult. For example, the Lock-It device, which was used as a fixation technique in the control group in the studies by Sharma et al. and Viburajah et al., resulted in notably lower incidences of migration in both (0 and 6%, respectively) compared to the other control groups [8,21]. We sought to solve this with a subgroup analysis of studies with tunneling and the control groups with adhesive dressing, without a suture or Lock-It device, only. In this subgroup analysis, a positive effect of tunneling on migration only remained significant for inward migration.

Secondly, the outcome was influenced by the timepoints chosen and the cut-off values and definitions employed. To address these limitations, a subgroup analysis was performed first by pooling only those studies in line with the predefined definition of clinically relevant migration. The duration of epidural treatment in some studies was relatively short. This could have led to an underestimation of the true incidence of infection, as the risk of infection increases with prolonged duration.

Thirdly, there were some limitations in the review process. Not all studies (n = 5) were available for assessment; however, these studies were mainly performed before 1995 and did not include randomized controlled trials, registries, or cohort studies.

This is the first systematic review and meta-analysis on the effect of tunneling as a fixation method for epidural catheters on migration, infection, and analgetic effect. All studies on this topic were included to not miss any important data, and meta-analyses were performed with RCTs and large registries to gain the best quality of evidence.

It is unlikely that any specific fixation technique will completely eliminate catheter migration. Although a significant benefit of tunneling in the prevention of inward migration was found, we doubt its clinical relevance. Therefore, possible benefits of tunneling must be weighed against the advantages of conventional fixation techniques, for instance, more operator-friendly, less time-consuming, more comfortable for the patient, and perhaps more cost-effective [7,8]. Furthermore, the additional needle punctures of tunneling may even do harm, with higher incidences of erythema and bleeding complications [8]. We would therefore advise considering tunneling based on an individualized benefit–risk analysis. For example, for obese patients or patients requiring prolonged epidural analgesia, where adhesive tape can peel off with friction, tunneling might be considered. However, based on our results, we do not recommend tunneling for routine perioperative practice, since there is a lack of benefit and a lack of difference in outcome. Future studies should focus on more modern fixation devices, like the Lock-It device. Preliminary results appear to be promising based on the data of the included studies in this review.

## 5. Conclusions

In conclusion, the results of this analysis suggest that tunneling of the epidural catheter reduces the risk of inward migration, albeit with questionable clinical relevance, but not outward migration. Furthermore, tunneling does not reduce the risk of infection. Pain scores, epidural failure, or patient satisfaction seem not to be relevantly affected by the fixation technique.

## Figures and Tables

**Figure 1 jcm-14-05788-f001:**
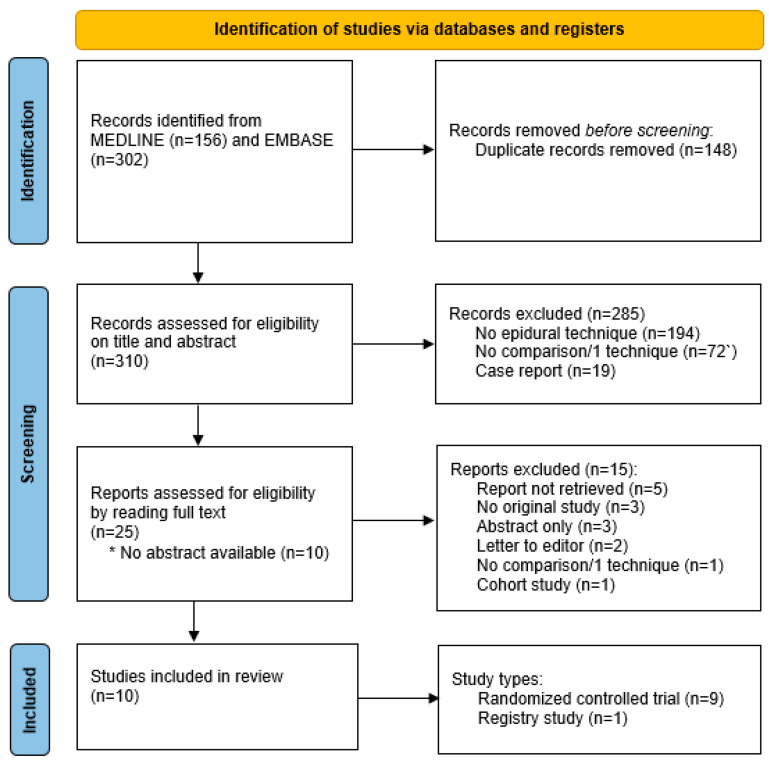
Flowchart of the systematic review and search strategy.

**Figure 2 jcm-14-05788-f002:**
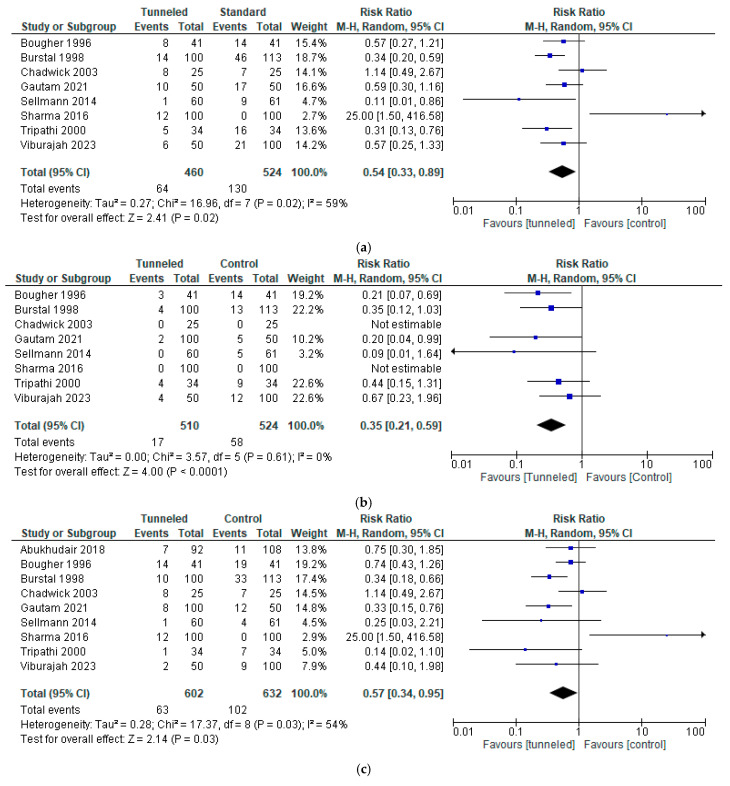
Forest plot of the incidence of migration in the tunneled vs. the conventional fixation method group [3,8,13,14,17,18,19,20,21]: (**a**) any form of migration; (**b**) inward migration; (**c**) outward migration.

**Figure 3 jcm-14-05788-f003:**
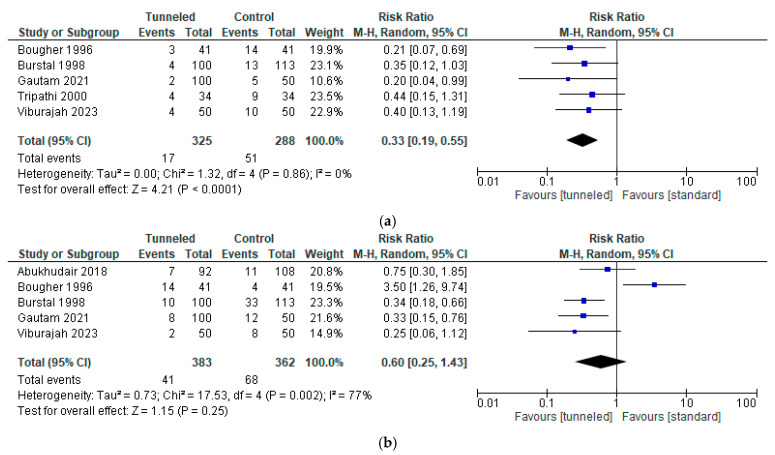
Forest plot of the incidence of migration in the tunneled vs. the conventional fixation method group (only tunneling vs. adhesive dressing (no sutures or Lock-It)); (**a**) inward migration [3,13,18,20,21]; (**b**) outward migration [3,13,17,18,21].

**Figure 4 jcm-14-05788-f004:**
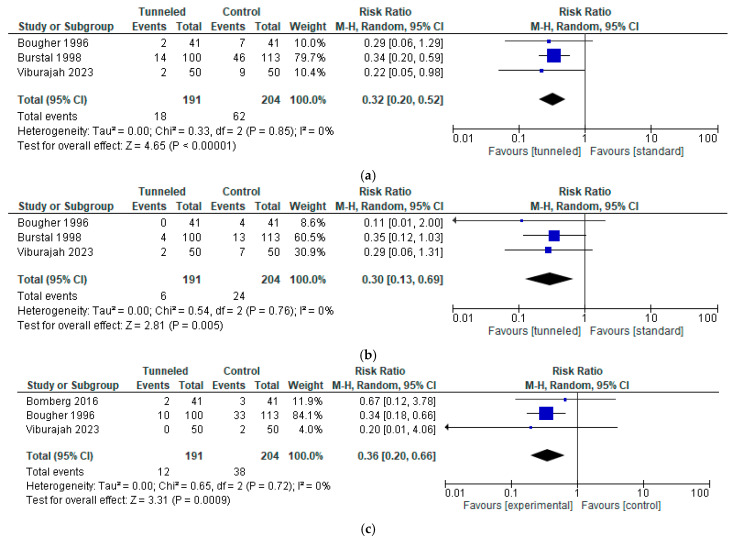
Forest Plot of the incidence of significant migration (>1 cm inwards or >2 cm outwards) in the tunneled vs. the conventional fixation method group (only tunneling vs. adhesive dressing (no sutures or Lock-It device)): (**a**) any form of migration [13,18,21]; (**b**) inward migration [13,18,21]; (**c**) outward migration [9,18,21].

**Figure 5 jcm-14-05788-f005:**
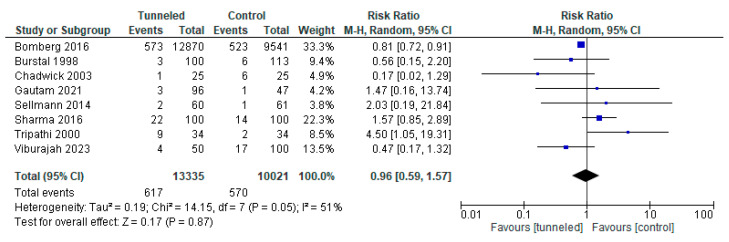
Forest plot of the incidence of infection in the tunneled vs. the conventional fixation method group [3,8,9,13,14,19,20,21].

**Table 1 jcm-14-05788-t001:** Characteristics of the included studies.

Author, Date	Type	Population	Level Epidural	T/Co	Tunneling Technique	Fixation Technique Control	Tunneled Mean Duration Epidural (SD)	Control Mean Duration Epidural (SD)
Abukhudair 2018 [17]	RCT	>18 years, major surgery, ASA 1 or 2	Both; 94%/6% ^1^	92/108	No loop, Tuohy needle, 5 cm lateral	Adhesive tape	2.7 d (range 1–4 d)	2.5 d (range 1–5 d)
Bomberg 2016 [9]	Registry	All thoracic epidural analgesia included in the German Network for Regional Anesthesia registry (25 hospitals)	Thoracic	12,870/9541	N.A.	N.A.	6.0 d (2.4 d)	5.7d (2.2 d)
Bougher 1996 [18]	RCT	Post-operative epidural analgesia	Both; 45%/55% ^1^	41/41	No loop, Tuohy needle, 5 cm lateral	Loop, adhesive spray, transparent adhesive dressing	2.20 d (0.87 d)	2.22d (1.11 d)
Burstal 1998 [13]	RCT	Post-operative epidural analgesia > 2 days	Both; 83%/17% ^1^	100/113	No loop, 16G venflon, 6–8 cm lateral	N.A.	3.5 d (1.3 d)	3.1d (1.5 d)
Chadwick 2003 [19]	RCT	Mentally competent, major abdominal surgery	Thoracic; Th7–Th12	25/25	No loop, 14G needle, 5 cm	Adhesive foam + suture, tegaderm	4.7 d (1.4 d)	5.3 d (1.7 d)
Gautam 2020 [3] ^2^	RCT	20–65 years, major upper abdominal surgery, ASA I/II	Thoracic; Th7–Th10	1002/50	TG 1; with loop (1.0 cm), 16G venflon, 4 cm cranial; TG 2 no loop, 16G venflon, 4 cm cranial	Transparent adhesive dressing	TG1 5.1 d (1.4 d); TG2 5.9 d (1.2 d)	5.7 d [1.3 d]
Sellman 2014 [14]	RCT	>18 years, major abdominal or thoracic surgery	Thoracic; Th3–Th12	60/61	No loop, 16G venflon, >2 cm lateral, suture	Steri-strips, tegaderm	109 h (46 h)	97 h (37 h)
Sharma 2016 [8]	RCT	All, lower extremity orthopedic surgery	Lumbar; L3–L5	100/100	With loop (1.5 cm), Tuohy needle, 3–5 cm caudal/cranial	Lock-It device	72 h	72 h
Tripathi 2000 [20]	RCT	Post-operative thoracic epidural analgesia	Thoracic; Th7–Th11	34/34	With loop (1.5 cm), Tuohy needle, 3–5 cm caudal	Loop, adhesive dressing	2.4 d (3.0 d)	4.6 d (2.1 d)
Viburajah 2023 [21] ^3^	RCT	Adult, non-obstetric surgery patients under regional anesthesia with a lumbar epidural	Lumbar; L2–L4	50/1003	No loop, Tuohy needle, 2–3 cm caudal	Co1: transparent adhesive dressing tape; Co2: Lock-It device	48 h	48 h

RCT = randomized controlled trial, Th = thoracic level, L = lumbar level, T = tunneled, Co = control, d = days, h = hours, SD = standard deviation. ^1^ Percentages (xx%/xx%) represent the percentage of thoracic vs. lumbar epidural catheters in this study. ^2^ The study of Gautam et al. [3] included two groups (both n = 50) with different tunneling techniques, as specified in the column “Tunneling technique”. ^3^ The study of Viburajah et al. [21] included two groups with different conventional fixation techniques, as specified in the column “Fixation technique control”.

**Table 2 jcm-14-05788-t002:** Definitions of outcomes of the included studies.

Study	Definition Migration	Definition Infection	Definition Analgesia
Abukhudair 2018 [17]	Found loose, entirely outside patient, leaking	Not studied	Patient’s complaints of uncontrolled pain; Faces Pain scale (rest/dynamic)
Bomberg 2016 [9]	Not studied	Classification German Society of Anesthesiologists ^1^	Not predefined: In results, pain intensity, rest/activity, infusion rate, patient satisfaction
Bougher 1996 [18]	Any in- or outward migration	Not studied	Not studied
Burstal 1998 [13]	Inward > 1 cm; Outward > 2.5 cm	Erythema and induration > 5 mm around the skin exit site; visible pus	Not studied
Chadwick 2003 [19]	Inward > 1 cm; Outward > 2 cm	Any signs of inflammation at the site	Patient opinion on technique and quality of analgesia; analgesic failure requires alternative analgesia or replacement epidural
Gautam 2020 [3]	Inward ≥ 1 cm; Outward ≥ 2 cm	Erythema and induration > 5 mm around the skin exit site; visible pus	Analgetic adequacy; inadequate analgesia requires alternative analgesia or replacement epidural
Sellman 2014 [14]	Inward > 2 cm; Outward > 2 cm	Classification German Society of Anesthesiologists ^1^	NRS at rest/activity, after removal, patient satisfaction
Sharma 2016 [8]	Inward > 1 cm; Outward > 1 cm	Erythema; induration	VAS pain scores + patient’s comfort during epidural fixation by Likert’s score
Tripathi 2000 [20]	Any in- or outward migration	Local inflammation	Not studied
Viburajah 2023 [21]	Any in- or outward migration	Erythema, pain, and induration > 5 mm around the skin exit site; visible pus	Analgetic adequacy, pain at the site of injection, patient satisfaction scores (1–5)

^1^ Classification of infection as recommended by the German Society of Anesthesiologists: mild, moderate, and severe catheter-related infections: (i) mild infections = at least two out of three infection signs (redness, swelling, or local pain); (ii) moderate infections = mild plus at least one of the following findings: increased C-reactive protein, leukocytosis, fever, or pus at the punctured site; and (iii) severe infections = the need for a surgical incision or revision.

**Table 3 jcm-14-05788-t003:** Result of analgetic effect, pain scores, or patient satisfaction between the tunneled and conventional fixation method groups.

Study	Analgesia Overall
Abukhudair 2018 [17]	63/111 uncontrolled pain; No difference in uncontrolled pain between groups (*p* = 0.282).
Bomberg 2016 [9]	Only ± 50% of data available; Less pain in rest and during activity at 24 h (*p* < 0.001) in the tunneled group; Higher patient satisfaction overall (*p* < 0.001) in the tunneled group
Burstal 1998 [13]	159 catheters functional at time removal; 83% tunneled; 67% control; *p* = 0.008. Functional without significant movement; 77% tunneled vs. 50% control; *p* < 0.0001. Functional > 2 days; 68% tunneled; 54% standard *p* = 0.037.
Chadwick 2003 [19]	2/6 patients in the tunneled group with > 4 cm migration not satisfied with analgesia; 1 patient in the tunneled group unsatisfied with analgesia at day 2 without movement; 1 patient in the control group (sutured) unsatisfied with analgesia < 12 h after placement.
Gautam 2020 [3]	8/96 (8%) patients of the tunneled and 8/47 (17%) patients of the control group experienced failure of analgesia. Of these 16 patients, catheter migration was observed in 8.
Sellman 2014 [14]	No difference in NRS during and after study (*p* = 0.26)
Sharma 2016 [8]	77% patients in the tunneled group disliked tunneling as a method
Viburajah 2023 [21]	No clinically significant difference in VAS scores between the groups; Significant difference in satisfaction in favor of the tunneled and control with Lock-It group, compared to the control with adhesive dressing conventional and other groups (*p* < 0.001).

## Data Availability

All data used in this study are available in the Appendix A section.

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
