# Peer review of "Prevention of Epidural Catheter Migration and Inflammation by Tunneling: A Systematic Review and Meta-Analysis"

_jcm, 2025, doi:10.3390/jcm14165788_

Round 1
Reviewer 1 Report
Comments and Suggestions for Authors
I have reviewed the manuscript titled “Prevention of epidural catheter migration and inflammation by tunnelling: a Systematic Review and Meta-Analysis”. I thank the authors and the editors for the opportunity to review this well-written paper. Authored by an experienced group of researchers, the manuscript presents the findings of a reasonably well-designed systematic review and meta-analysis assessing the efficacy of epidural catheter tunnelling in catheter infection or migration, compared to the conventional fixation technique.
Although I was unfortunately unable to access the supplementary materials of this manuscript, I commend its strong conceptual foundation, as well as the clarity and professionalism reflected in its composition and presentation. The results are accurately presented, with subgroup analysis - differentiating migration outcomes by fixation technique – being well-argued and thoroughly discussed.
The Discussion section includes pertinent interpretation of the results and appropriately acknowledges the study’s limitations. The authors also offer their valuable clinical perspective derived from the results in the last paragraph of the section.
Overall, the methodology is sound, the analytical approaches are appropriate, and I have no significant concerns regarding the validity of the findings.
Author Response
Comments 1: I have reviewed the manuscript titled “Prevention of epidural catheter migration and inflammation by tunnelling: a Systematic Review and Meta-Analysis”. I thank the authors and the editors for the opportunity to review this well-written paper. Authored by an experienced group of researchers, the manuscript presents the findings of a reasonably well-designed systematic review and meta-analysis assessing the efficacy of epidural catheter tunnelling in catheter infection or migration, compared to the conventional fixation technique. Although I was unfortunately unable to access the supplementary materials of this manuscript, I commend its strong conceptual foundation, as well as the clarity and professionalism reflected in its composition and presentation. The results are accurately presented, with subgroup analysis - differentiating migration outcomes by fixation technique – being well-argued and thoroughly discussed. The Discussion section includes pertinent interpretation of the results and appropriately acknowledges the study’s limitations. The authors also offer their valuable clinical perspective derived from the results in the last paragraph of the section. Overall, the methodology is sound, the analytical approaches are appropriate, and I have no significant concerns regarding the validity of the findings.
Response 1: Thank you very much for taking the time to review this manuscript. We are sorry to hear that you were unable to access the supplementary materials, and we hope that this problem will be solved before final publication. We have attached the supplementary materials as a PDF fille below (unfortunately not in the best quality due to the Excel to PDF conversion).

Reviewer 2 Report
Comments and Suggestions for Authors
The prevention of epidural catheter migration and local or systemic inflammation respectively are an ongoing discussion in anesthesia. Since decades researchers try to find evidence and identify the holy grail of securing the epidural catheter in situ in order to prevent secondary misplacement with a loss of function, either pain – migration out of the epidural compartment or a more or a total neuro-axial sensoric and motoric block – migration through and perforating the dura mater. Hence, reduction of mechanical friction and stress on the epidural catheter was hypothesized to be as well related to reduced local or systemic inflammation. In the recent German S1 (July 2025) guideline on Hygiene recommendations for regional anesthesia authors found no evidence for a reduction of infection via tunneling but recommended tunneling for catheters with an expected duration of therapy > 72hrs an increased individual risk of infection (Anaesthesiologie. 2025 Aug;74(8):504-515. German. doi: 10.1007/s00101-025-01563-0. Epub 2025 Jul 23. PMID: 40702337). To exclude the maximum bias the authors have chosen for a model of studies comparing any epidural tunnel fixation technique with any conventional epidural fixation technique, with exclusion of the obstetric population, and the exclusion of caudal anesthesia, due to their special stress, tissue and environmental/hygenic profiles.In total 11 studies, with a number of 23,695 patients, have been included. Pooled data of the nine studies reporting data on migration showed that tunneling reduced the incidence of inward migration (RR, 0.33; 95% CI, 0.19 to 0.55; n = 613; p<0.0001; I2 = 0%), but not outward migration (RR, 0.60; 95% CI, 0.25 to 1.43; n = 745; p = 0.25; I2 = 77%). Pooled data from the nine studies on the incidence of infection revealed no statistically significant group difference, what goes along with the previously cited German S1 guideline.
In Summary the authors died well in selecting relevant studies and excluding weak variables like delivery and caudal anesthesia. Moreover, only trials with a direct comparison, intervention vs control, have been included. Addressing strength and limitations the authors underlined the variety of fixating catheters, that might have influenced findings concerning migration and inflammation.
Author Response
Comment 1: In the recent German S1 (July 2025) guideline on Hygiene recommendations for regional anesthesia authors found no evidence for a reduction of infection via tunneling but recommended tunneling for catheters with an expected duration of therapy > 72hrs an increased individual risk of infection (Anaesthesiologie. 2025 Aug;74(8):504-515. German. doi: 10.1007/s00101-025-01563-0. Epub 2025 Jul 23. PMID: 40702337). To exclude the maximum bias the authors have chosen for a model of studies comparing any epidural tunnel fixation technique with any conventional epidural fixation technique, with exclusion of the obstetric population, and the exclusion of caudal anesthesia, due to their special stress, tissue and environmental/hygenic profiles.In total 11 studies, with a number of 23,695 patients, have been included. Pooled data of the nine studies reporting data on migration showed that tunneling reduced the incidence of inward migration (RR, 0.33; 95% CI, 0.19 to 0.55; n = 613; p<0.0001; I2 = 0%), but not outward migration (RR, 0.60; 95% CI, 0.25 to 1.43; n = 745; p = 0.25; I2 = 77%). Pooled data from the nine studies on the incidence of infection revealed no statistically significant group difference, what goes along with the previously cited German S1 guideline.
In Summary the authors died well in selecting relevant studies and excluding weak variables like delivery and caudal anesthesia. Moreover, only trials with a direct comparison, intervention vs control, have been included. Addressing strength and limitations the authors underlined the variety of fixating catheters, that might have influenced findings concerning migration and inflammation.
Response 1: Thank you very much for taking the time to review this manuscript. Thank you for pointing out the recent German S1 (July 2025) guideline on Hygiene recommendations for regional anesthesia, we are pleased to see that there conclusion is in line with our conclusion.

Reviewer 3 Report
Comments and Suggestions for Authors
The authors conducted a meta-analysis comparing two methods of securing epidural catheters: conventional and tunneling. Both in the abstract and in the introduction, the authors need to explain what is meant by conventional. Also, the pronouns "we" and "our" should be removed throughout the manuscript.
Specific Comments:
Line 35: Change "epidurals" to "epidural catheters"
Line 39: What is the incidence of outward and inward migration of epidural catheters with conventional techniques?
Line 41: Please explain what complications occur from inward migration.
Line 275: Is this the most common route of bacterial infection? How does it compare to improper skin preparation or infection with needle insertion?
Line 318: Suggest deleting, "However, to our best knowledge"
Line 330: Please explain why you are recommending tunneling given the lack of benefit and the lack of difference in outcome.
Author Response
Comment 1: The authors conducted a meta-analysis comparing two methods of securing epidural catheters: conventional and tunneling. Both in the abstract and in the introduction, the authors need to explain what is meant by conventional.
Response 1: We have added the following sentences to explain the conventional method of securing an epidural catheter “(any technique, e.g. adhesive tape)”. This explanation is in line with the more extensive explanation in the method section.
Comment 2: Also, the pronouns "we" and "our" should be removed throughout the manuscript.
Response 2: Done. However when it concerns our opinion we have still used “we/our” (in the discussion section)
Comment 3: Line 35: Change "epidurals" to "epidural catheters"
Response 3: Done
Comment 4: Line 39: What is the incidence of outward and inward migration of epidural catheters with conventional techniques?
Response 4: The incidence (between 5% and 50%) mentioned in Line 39 is the incidence of any migration (in or outward) with any conventional technique mentioned in previous literature. The incidences of in- and outward migration of epidural catheters fixed by conventional techniques ranged from 0-81% with a mean of 28.4% in our review (versus 0-42% and a mean of 15.9% in the tunneled group). If we divide the conventional group in inward or outward migration, the mean incidence for inward migration was 12.9% (range 0-34.1%) and 19.2% (range 0-46.3%) for outward migration (versus 3.6% [range 0-11.8%] and 13.7% [range 1.7-34.1%] in the tunneled group). Because the latter results are from of our review we only provide the incidence described in previous literature in the introduction section.
Comment 5: Line 41: Please explain what complications occur from inward migration.
Response 5: We have added the following sentence to describe potential complications of inward migration “Catheter migration can lead to loss of analgesic effect, especially when migrating outward or laterally (transforaminal escape), and to potential complications, such as intravascular, subdural or subarachnoid injection of local anesthetic, if migration is directed inwards”
Comment 6: Line 275: Is this the most common route of bacterial infection? How does it compare to improper skin preparation or infection with needle insertion?
Response 6: Yes, the most common route of bacterial infection of the epidural catheter as described in literature is via the cutaneous track. Although there are several ways an infection can occur: by contamination of the exit site and subsequent spread along the catheter track, by organisms being introduced during needle or catheter insertion or by haematogenous spread to the site from the blood-stream or from a distant focus of infection. The first one seems to be the most important one as studies show that most of the microorganisms cultured from epidural catheters are members of the normal skin flora.
We have adjusted the sentence to “The most common route of bacterial pathogen migration is contamination of the exit site and subsequent spread along the catheter track”, to make this more clear.
Comment 7: Line 318: Suggest deleting, "However, to our best knowledge"
Response 7: Done
Comment 8: Line 330: Please explain why you are recommending tunneling given the lack of benefit and the lack of difference in outcome.
Response 8: We agree that based on our results there is lack of benefit and lack of difference in outcome, therefore we do not recommend tunneling of the epidural catheter. To emphasize this we have slightly adjusted our conclusion “However, for routine perioperative practice, tunneling cannot be recommended based on our results.” To “ However, based on our results we do not recommend tunneling for routine perioperative practice, since there is lack of benefit and lack of difference in outcome.”
